# Hydroxyapatite Growth on Activated Carbon Surface for Methylene Blue Adsorption: Effect of Oxidation Time and CaSiO₃ Addition on Hydrothermal Incubation

**Anastasio Moreno-Santos [1], Jorge Carlos Rios-Hurtado [1,*], Sergio Enrique Flores-Villaseñor [1], Alma Graciela Esmeralda-Gomez [2], Juanita Yazmin Guevara-Chavez [1], Fatima Pamela Lara-Castillo [1] and Griselda Berenice Escalante-Ibarra [1]**

[1] Facultad de Metalurgia, Universidad Autonoma de Coahuila, Carretera 57 Km 5, Los Bosques, Monclova 25710, Mexico

[2] Facultad de Ingeniería Mecánica y Eléctrica, Universidad Autonoma de Coahuila, Carretera Torreon-Matamoros Km 7.5, Ciudad Universitaria, Torreon 27410, Mexico

[*] Correspondence: jorgerios@uadec.edu.mx

**Featured Application: Adsorption technology for organic pollutants with new adsorbents materials.**

**Abstract:** Many adsorbent materials are now commercially available; however, studies have focused on modifying them to enhance their properties. In this study, an activated carbon (AC) and hydroxyapatite (HAp) composite was synthesized by the immersion of ACs in a simulated body fluid solution, varying the AC oxidation degree along with the addition of CaSiO₃. The resulting composites were characterized by ash %, X-ray fluorescence (XRF), Fourier-transformed infrared spectroscopy (FTIR), scanning electron microscopy (SEM), and point of zero charge (PZC). The characterization results indicated that the addition of CaSiO₃ and the oxygenated functional groups in the AC surface are key factors for HAp growth. The composites were tested on methylene blue (MB) adsorption as a potential application for the synthesized materials. Adsorption isotherms were modeled with Langmuir and Freundlich isotherms, and the composites were fitted to a Langmuir model with the highest $q_{max}$ value of 9.82. The kinetic results indicated that for the pseudo-second-order model, the composites fitted, with a contact time of 180 min to remove a 95.61% average of the MB. The results indicate that composite materials can be an efficient adsorbent for the removal of MB from aqueous solutions at low concentrations since the material with the highest amount of HAp growth removed 99.8% of the MB in 180 min.

**Keywords:** activated carbon; CaSiO₃; oxidation; hydroxyapatite; methylene blue; adsorption

## 1. Introduction

The use of hydroxyapatite (HAp) with carbon materials has become very common nowadays, as it can be considered a novel material that can be used in diverse applications. K. Pereira et al., in 2019, obtained oxidized carbon nanotubes modified with HAp particles that showed great potential for bone regeneration applications [1]. Q. Liu et al., in 2018, obtained a bonding between activated carbon fibers (ACFs) with HAp by electrochemical deposition, resulting in an ACF/HAp composite to be used as a biofilm for support in bioreactors because of its strong ability to immobilize activated sludge and bacteria [2]. Jayaweera et al., in 2018, conducted studies with microorganisms on HAp-modified carbon-based materials demonstrating antimicrobial and metal ion removal properties [3].

The combination of carbon-based materials with HAp has also been widely studied in the environmental field, where the most frequent application is the removal of pollutants in aqueous solutions. M. Shanika et al., in 2015, studied the adsorption of Pb²⁺ by a composite prepared with granular activated carbon and HAp [4]. Y. Lang et al., in 2019, synthesized

a granular activated carbon-hydroxyapatite composite by the chemical coprecipitation of $CaCl_2$ on the activated carbon in a $(NH_4)_2HPO_4$ solution for $Pb^{2+}$ removal [5]. Y. Zhu et al., in 2018, investigated the adsorption of $Pb^{2+}$ ions with a composite material obtained from the carbonized organic material of sugarcane with HAp [6].

These types of composite materials improve adsorption capacity due to the ion exchange properties of HAp and the carboxylic functional groups (-COOH) present in the carbon material surface; so, increasing these groups on the surface is of great importance. Several studies have focused on increasing the oxygenated functional groups. Ahmad et al., in 2020, showed the roles of the different surface functional groups in graphene-based nanomaterials [7]. Zhang et al., in 2018, prepared biochar from rice straw and modified the surface with the carboxyl and phenol groups for $Cd^{2+}$ adsorption [8]. Liang et al., 2018, studied the influence of functional groups on activated carbon (AC) and the effects of acetone adsorption [9]. Li et al., in 2022, investigated the effect of functional groups on VOCs' adsorption [10].

The adsorption of organic molecules by AC/HAp composites is one potential application. Ferri et al., 2021, reported the sorption of organic/inorganic molecules by hydroxyapatite/carbon composites [11]. Nurhadi et al., 2019, reported the adsorption of methylene blue (MB) by AC/HAp composites [12]; Yiming et al., 2018, reported the obtention of biochar-supported HAp and MB adsorption [13]. Yang et al., in 2015, obtained a magnetic carbon with HAp for the removal of organic compounds [14].

The purpose of this work was to obtain a composite of granular activated carbon with hydroxyapatite grown by the hydrothermal method, modifying parameters, such as the AC surface oxidation and the addition of $CaSiO_3$, to improve the adsorptive properties of the materials. The obtained materials were tested for the adsorption and kinetics removal of methylene blue as a potential application.

## 2. Materials and Methods

Activated carbon from coconut shell, PKCARBON1240® carbon, was commercially acquired from the "PURIKOR" company. Samples were washed several times with distilled water and dried at 80 °C for 24 h. The chemical reagents used in this study were: $CaCl_2$, KCl, $NaHCO_3$, NaCl, $MgCl_2*6H_2O$, $Na_2SO_4$, HCl, $HNO_3$ (CTR Scientific, Monterrey, Mexico) $K_2HPO_4$ (Fermont, Monterrey, Mexico), and $(CH_2OH)_3CNH_2$ (Sigma Aldrich, Monterrey, Mexico).

### 2.1. Activated Carbon Oxidation

Activated carbon oxidation was carried out according to the Rangel–Méndez and Streat process in 2006 [15]. Briefly, carbon and 8 M $HNO_3$ were placed in a reactor in a 1:4 mass:volume ratio at 80 °C. Different oxidation times of 1 and 2 h were tested. After this time, the reactor was immediately cooled, and the activated carbon was decanted and rinsed with distilled water until a pH of $6 \pm 0.1$ was reached. Finally, the materials were dried at 90 °C for 24 h.

### 2.2. HAP Growth on Carbon Surface

Surface modification was performed as follows: 0.1 g of each oxidized carbon material, 20 mg of $CaSiO_3$, and 40 mL of a simulated body fluid (SBF) were placed in 50 mL polypropylene conical tubes and incubated for 21 days at 36.5 °C. The SBF solution was prepared according to the experiments of Kokubo and Takadama in 2006 [16]. After the incubation time, the materials were washed with distilled water and dried at 90 °C for 24 h. A control material with no $CaSiO_3$ was also evaluated.

### 2.3. Methylene Blue Adsorption Experiments

20 mg of the resulting materials and 20 mL of a 1 ppm methylene blue (MB) solution were placed in a 50 mL conical tube. The pH was adjusted to $7 \pm 0.002$ in an Orion StarA2110 potentiometer using 0.1 M NaOH or 0.1 M $HNO_3$. Samples were incubated

at 25 °C, and the pH was measured over periods of 12 hours. Once the equilibrium was reached (48 h), the remaining MB concentration was determined by UV-Vis spectrometry using a wavelength of 664 nm.

The amount of MB adsorbed ($C_e$) and the adsorption capacity ($q_e$) of the dye were calculated by mass balance and Equation (1), respectively, to obtain the adsorption isotherms. Finally, the isotherms were fitted to the Langmuir and Freundlich models using the equations (Equations (2) and (3)).

$$q_e = \frac{(C_o - C_e)V}{m} \tag{1}$$

$$q_e = \frac{q_{max}k_L C_e}{1 + k_L C_e} \tag{2}$$

$$q_e = k_F C_e^{1/n} \tag{3}$$

$q_e$ = Adsorption capacity at equilibrium, mg·g$^{-1}$
$C_o$ = Initial concentration of the solution, mg·L$^{-1}$
$C_e$ = Equilibrium concentration of the solution, mg·L$^{-1}$
$q_{max}$ = Maximal adsorption capacity, mg·g$^{-1}$
$k_L$ = Adsorption energy, L·mg$^{-1}$
$k_F$ = Adsorption capacity index, mg·g$^{-1}$ (L·mg$^{-1}$)$^{1/n}$
$n$ = Ratio of adsorption intensity

*2.4. Methylene Blue Adsorption Kinetics*

For the material, 60 mg of each, and 60 mL of a 1 ppm MB solution, were placed in a 100 mL beaker and stirred (250 rpm) at room temperature, and the pH was adjusted to $7.00 \pm 0.002$ in an Orion StarA2110 potentiometer. Different times were evaluated: 1, 3, 5, 10, 20, 40, 40, 80, 120, 160, 180, 220, 240, 280, and 300 min. After each time, the MB concentration was measured with UV-Vis spectroscopy (664 nm).

Subsequently, the kinetic model was calculated using Equations (4) and (5), which represent the pseudo-first-order and pseudo-second-kinetic processes. These experiments were performed in triplicate.

$$q_t = q_e \left(1 - e^{-k_1 t}\right) \tag{4}$$

$$q_t = \frac{(q_e^2 k_2 t)}{(1 + q_e k_2 t)} \tag{5}$$

$q_t$ = Amount of species adsorbed at time t, mg·g$^{-1}$
$q_e$ = Adsorption capacity at equilibrium, mg·g$^{-1}$
$k_1$ = rate constant (pseudo-first-order), min$^{-1}$
$k_2$ = rate constant (pseudo-second-order), g·mg$^{-1}$ min$^{-1}$
$t$ = Time, min

*2.5. Materials Characterization*
Ash Content

Each previously dried sample, at 0.1 g, was weighed three times on a tared crucible and placed in an oven for 16 h at 650 °C; after this time, the samples were removed, allowed to cool, and weighed at room temperature. The ash content was determined using Equation (6).

$$Ash = \frac{m_{Ash}}{m_{Sam}} \times 100 \tag{6}$$

*Ash* = Ash content, %
$m_{Ash}$ = Ash mass, g
$m_{Sam}$ = Dry sample mass, g

### 2.6. X-ray Fluorescence Spectroscopy (XRF)

X-ray fluorescence spectroscopy was carried out using a Malvern Panalytical model, Epsilon 1. The samples were sieved to reach the mesh size required by the equipment and to avoid heterogeneity.

### 2.7. Fourier-Transformed Infrared Spectroscopy (FTIR)

Fourier-transformed infrared spectroscopy was performed on a PerkinElmer Frontier model. The spectra of the samples were taken on a KBr type disc, in the range of 4500–400 $cm^{-1}$, using 32 scans and 4 $cm^{-1}$ resolution.

### 2.8. Scanning Electron Microscopy (SEM)

Scanning electron microscopy was performed using a Hitachi SU8230 microscope, where the images obtained were generated with a 15 kV voltage using a conventional slide and silver paint to retain the samples.

### 2.9. Charge Distribution and Point of Zero Charge

The point of zero charge (PZC) determination was carried out in 50 mL conical tubes where 0.1 g of each material was placed in contact with 10 mL of distilled water, and the pH was adjusted between 3 and 11 using NaOH and HCl 0.1 M. The samples were agitated for 48 h, and the samples' final pH was measured.

## 3. Results and Discussion

For a simple nomenclature, we used AC: activated carbon; Xh: where X is the oxidized time; 21: 21, for the days of incubation without $CaSiO_3$; HAp: for the materials incubated for 21 days with $CaSiO_3$.

### 3.1. Chemical Properties of the Surface Samples

For the commercial-activated carbon, 2.00% ($\pm$0.17) of the ash content was observed. For the oxidized materials by chemical modification at varying times, the AC-1h showed 1.40% ($\pm$0.16), and for the AC-2h, it was 1.00% ($\pm$0.14) (Figure 1). The decrease in the ash percentage was due to the treatment with $HNO_3$, which eliminated part of the ash in the unoxidized carbon, as mentioned by Lamaming J. et al. in 2022 [17]. Moreover, the samples with 21 days of incubation in the absence of $CaSiO_3$ showed a slight increase in the ash percentage attributed to precipitates on the surface from the ions present in the SBF solution, with an ash content of 3.00 ($\pm$0.23) for AC-21, 3.21 ($\pm$0.25) for AC-1h21, and 3.41% ($\pm$0.32) for AC-2h21. Finally, the samples with $CaSiO_3$ (AC-HAp, AC-1hHAp, and AC-2hHAp) presented a very significant increase in the ash percentage value at 4.72% ($\pm$0.45), 7.15 ($\pm$0.45), and 11.53% ($\pm$0.42), respectively. The increased values depended on the formation of hydroxyapatite deposits on the surface of the materials. The inorganic materials remained in the crucible at the end of the analysis since the boiling point was higher than the temperature used for the ash determination [18–20].

To identify the presence of HAp in the samples, X-ray fluorescence was used. In Table 1, the elemental composition of the samples is presented, and the differences between the Ca and P percentages are analyzed.

**Table 1.** Compositions of carbon-based materials.

| Sample | Composition | | | | |
|--------|------|------|------|------|------|
|        | Si   | P    | K    | Ca   | Fe   |
| AC     | 0.58% | 0.13% | 0.75% | 1.09% | 0.81% |
| AC-21  | 0.46% | 0.35% | 0.47% | 2.78% | 1.05% |
| AC-HAp | 0.60% | 0.46% | 0.35% | 4.77% | 0.55% |

**Table 1.** *Cont.*

| Sample | Composition | | | | |
|---|---|---|---|---|---|
| | **Si** | **P** | **K** | **Ca** | **Fe** |
| AC-1h | 0.55% | 0.12% | 0.25% | 0.98% | 0.39% |
| AC-1h21 | 0.68% | 0.34% | 0.50% | 3.98% | 0.37% |
| AC-1hHAp | 0.52% | 0.35% | 0.69% | 5.37% | 0.29% |
| AC-2h | 0.44% | 0.13% | 0.16% | 0.89% | 0.31% |
| AC-2h21 | 0.49% | 0.37% | 0.38% | 4.01% | 0.32% |
| AC-2hHAp | 0.55% | 0.40% | 0.89% | 6.25% | 0.45% |

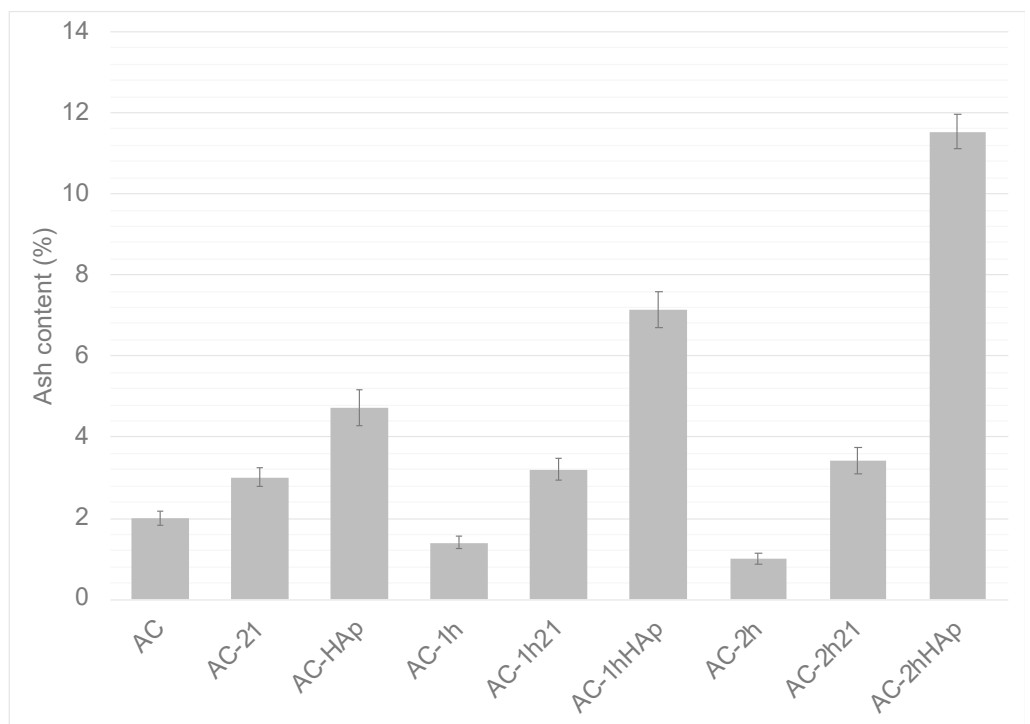

**Figure 1.** Ash content in percent of the different carbons tested (AC: activated carbon; 21: 21 days incubated; Xh: oxidized time; HAp: incubated for 21 days with $CaSiO_3$).

The commercial AC presented 1.09% Ca and 0.13% P due to the pyrolysis of the carbon precursor and activating agent ($H_3PO_4$). The materials incubated for 21 days, the AC-21, AC-1h21, and AC-2h21, showed an increase in the Ca and P percentages, with 2.78, 3.98, and 4.01% of the calcium and 0.35, 0.34, and 0.37% of the phosphorus, respectively. A higher increase in the percentage of Ca and P incubated with $CaSiO_3$ was observed, with 4.77, 5.37, and 6.25% for the Ca and 0.46, 0.35, and 0.40% for the P, corresponding to the AC-HAp, AC-1hHAp, and AC-2hHAp. The increased Ca and *p* values in these samples were due to the ion deposition in the incubation with the SBF solution [21–23], and the results demonstrated that, depending on the presence of $CaSiO_3$, an increase of Ca and P was observed. For the materials that were only oxidized, a reduction in the values of the Ca and P was observed due to the contact time with $HNO_3$ in the oxidation process, which removed the ash contained in the AC.

FTIR analysis identified the functional groups present in the adsorbent materials. In Figure 2, the IR spectra showed that all the adsorbent materials studied presented adsorption bands very similar to each other.

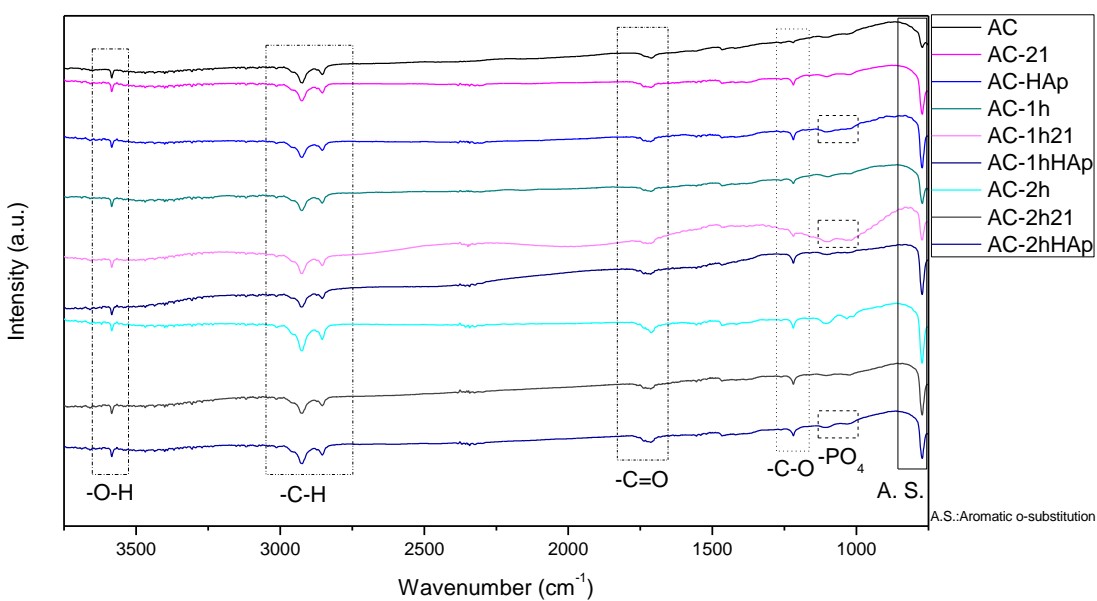

**Figure 2.** IR spectra of the carbon samples with different modifications.

The differences between the samples were the intensity of the signals. Initially, at 3584 cm$^{-1}$, a small and sharp peak was observed and corresponded to an O-H tension due to the hydroxyl that was present in the material since phenols were formed in the activation process [24]. The spectral bands around 2924–2853 cm$^{-1}$ belonged to the C-H stretching. Moreover, certain bands were appreciated between around 2400 and 2350 cm$^{-1}$, according to the C=O bond tension [25] and the overtones of the carboxylic groups, due to the oxidation process. Small signals at 1712 cm$^{-1}$ attributed to a band belonging to carbonyl stretching. In addition, all the materials had a very sharp band at about 1219 cm$^{-1}$ that was interpreted as a C-O strain, which increased considerably in the modified materials. Finally, at 775 cm$^{-1}$, they corresponded to an aromatic substitution of the ortho type by the formation of the compounds with C present in the aromatic group and another element or a bond with an active site. Small signals of the phosphate group were observed at 1116 and 1085 cm$^{-1}$, corresponding to HAp. An increase in the intensity of the AC-1h21 for the phosphate group signal can be attributed to the initial formation of HAp on the surface.

Moreover, a morphology analysis was carried out to corroborate Hap's existence. The morphology of the commercial-activated carbon (Figure 3) was a typical porous and irregular microstructure [24,26], with a particle size of 12–40 mm, presenting meso and macropores with an average pore diameter (p$_d$) of 34.00 nm and 1.38 μm, respectively.

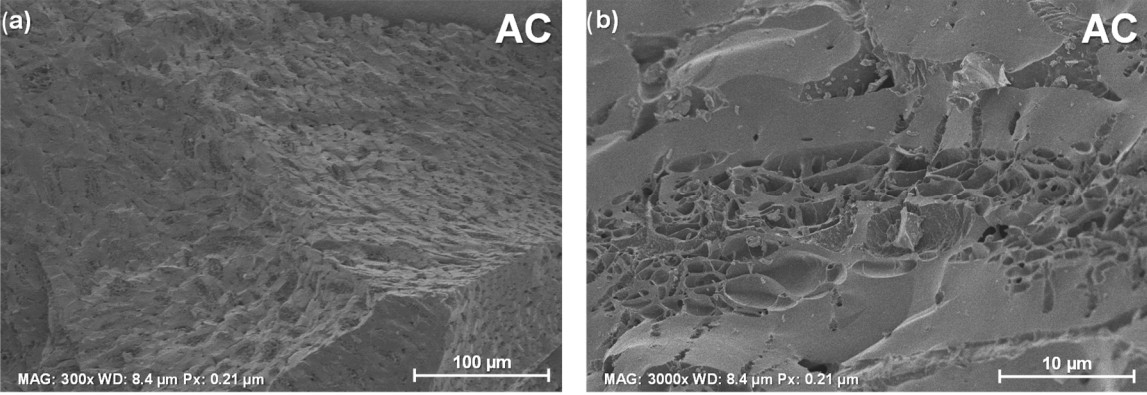

**Figure 3.** SEM images of the AC; (**a**) 300× and (**b**) 3000× magnification.

Figure 4 shows the micrographs of the oxidized materials. The 1 h (AC-1h) and the 2 h (AC-2h) oxidized carbon presented a change in the morphology, with a more irregular surface and an increase in the pore diameter; likewise, a skeletal morphology [27] in the material with a 2 h oxidation is observed. The change in morphology is directly related to the chemical oxidation with $HNO_3$ [28], as the acid in contact with the material modified the textural properties, increasing the surface porosity due to the carbon bonds' degradation.

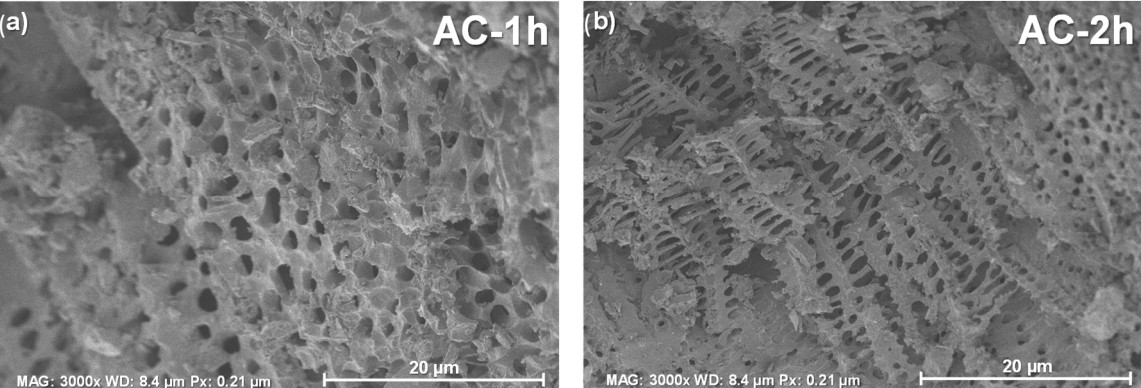

**Figure 4.** SEM images of the oxidized carbon samples; (**a**) AC-1h and (**b**) AC-2h.

For the samples incubated for 21 days, shown in Figure 5, typical HAp morphology was not observed; however, a dispersion of the ions and an initial growth of HAp were observed in a slight shining on the SEM images of the composite surface. The reason is that the $Ca^{2+}$ ions' concentration was insufficient for the spontaneous precipitation process to take place [22,23].

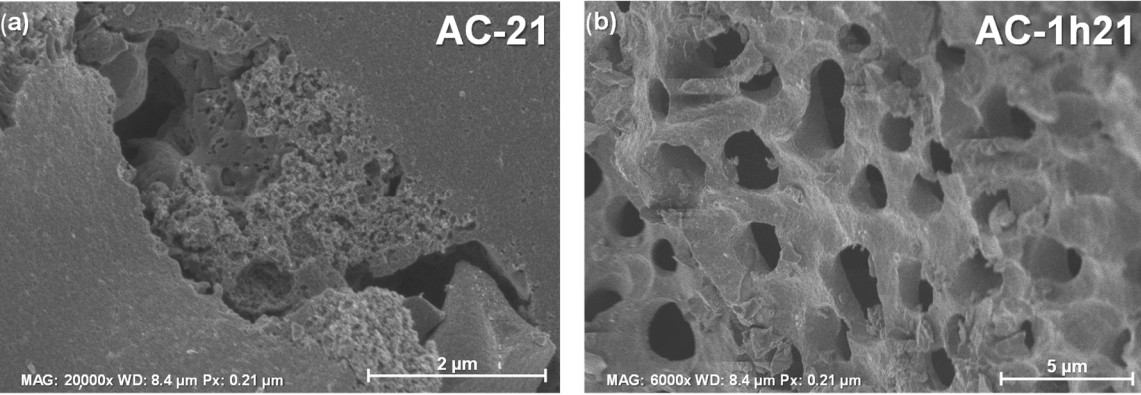

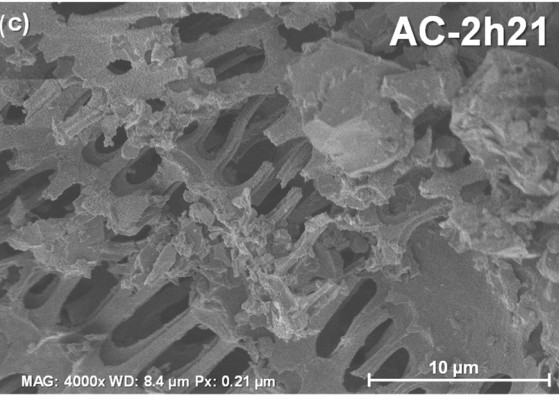

**Figure 5.** SEM images of the carbon samples incubated for 21 days; (**a**) AC-21, (**b**) AC-1h21, and (**c**) AC-2h21.

Furthermore, the samples incubated with CaSiO$_3$, as shown in Figure 6, showed a growth on the surface of the agglomerated deposits of the particles with a sphere-like morphology, attributed to the formation of the hydroxyapatite, with an average particle size of ~779.00 μm. The particle size varies due to the agglomeration, as particles of smaller than average size were visible (Figure 6c). The addition of CaSiO$_3$ during the incubation was a determinant factor for the formation of HAp on the substrate surface [29–31]. Calcium silicate powders transferred the ions to the SBF solution and adding Ca$^{2+}$ oversaturated the solution and successfully achieved the precipitation of hydroxyapatite on the surface [22]. In addition, there was a relationship between the oxidation degree and the growth of the hydroxyapatite particles on the carbon surface, explained by the fact that a higher concentration of the oxygenated functional groups on the carbon surface, derived from the modification with HNO$_3$, interacted with the calcium ions, and the precipitation of HAp was achieved.

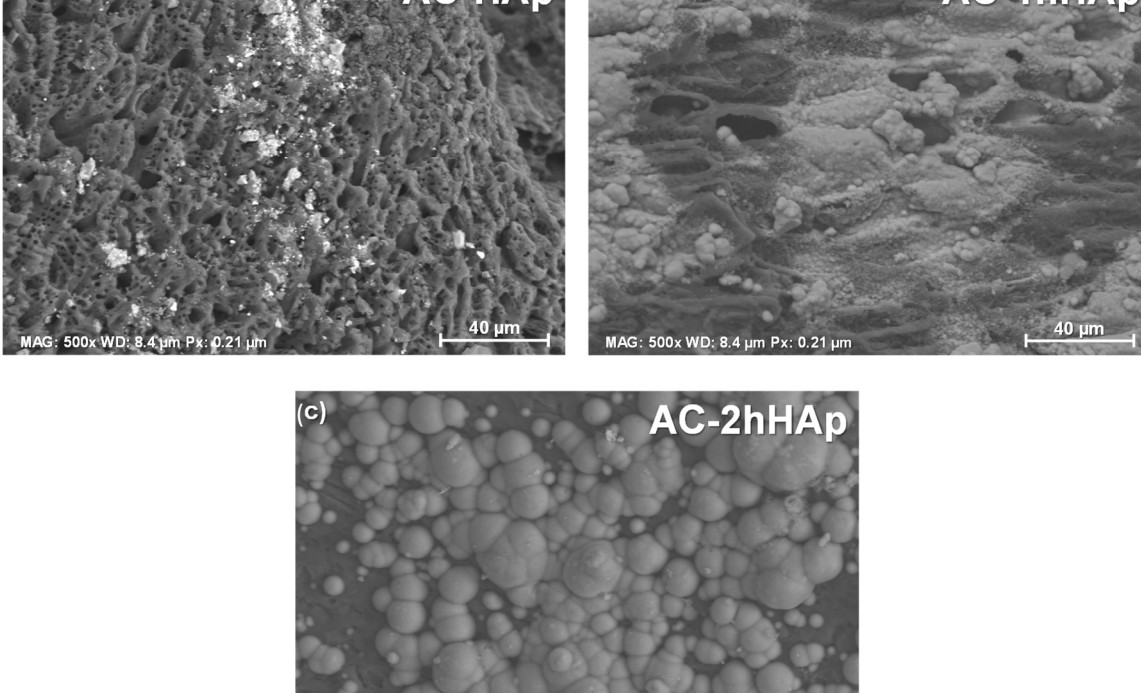

**Figure 6.** SEM microscopies of the HAp-modified carbon samples; (**a**) AC-Hap, (**b**) AC-1hHAp, and (**c**) AC-2hHAp.

To corroborate that the existing deposit on the surface corresponded to the HAp particles, an EDS analysis was performed. The chemical analysis of the AC (Figure 7a) had an 88 wt% of C, typical for these kinds of adsorbents; additionally, O and Si atoms were observed due to the activation process from the pyrolyzed coconut shell. The AC-HAp composition is presented in Figure 7; the AC-HAp, AC-1hHAp, and AC-2hHAp are visible in Figure 7b–d, and the presence of Ca and P atoms are observed, which the existence of these elements suggests that the AC was modified by the calcium phosphate shape, in addition to the results observed in the FTIR, XRF, and ash content [32].

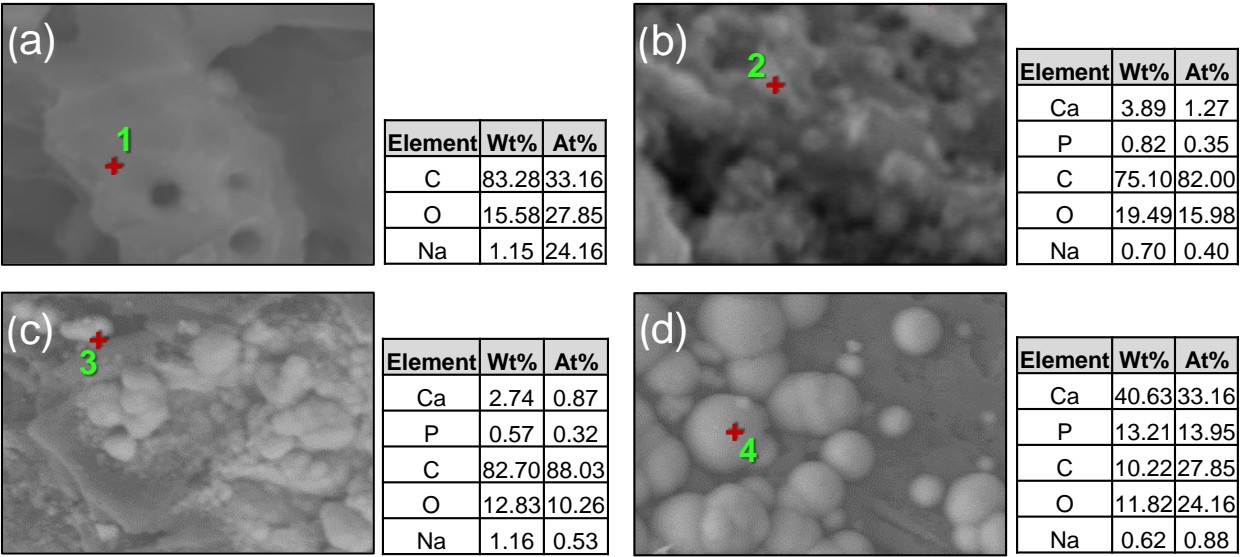

**Figure 7.** EDS analysis of carbon and HAp-modified carbon materials; (**a**) AC, (**b**) AC-HAp, (**c**) AC-1hHAp, and (**d**) AC-2hHAp.

### 3.2. Methylene Blue Adsorption Isotherms

To comprehend the adsorption mechanism on the samples, charge distribution experiments were carried out. Figure 8 shows the charge distribution of the materials studied, according to the method reported by Mahmood et al., 2011 [33]. The hydroxyapatite presented a basic behavior characteristic, with a PZC of 10.91, while the rest of the materials (the AC and its respective modifications) had a more complex character and a PZC that varied depending on the oxidation degree and the CaSiO$_3$ addition.

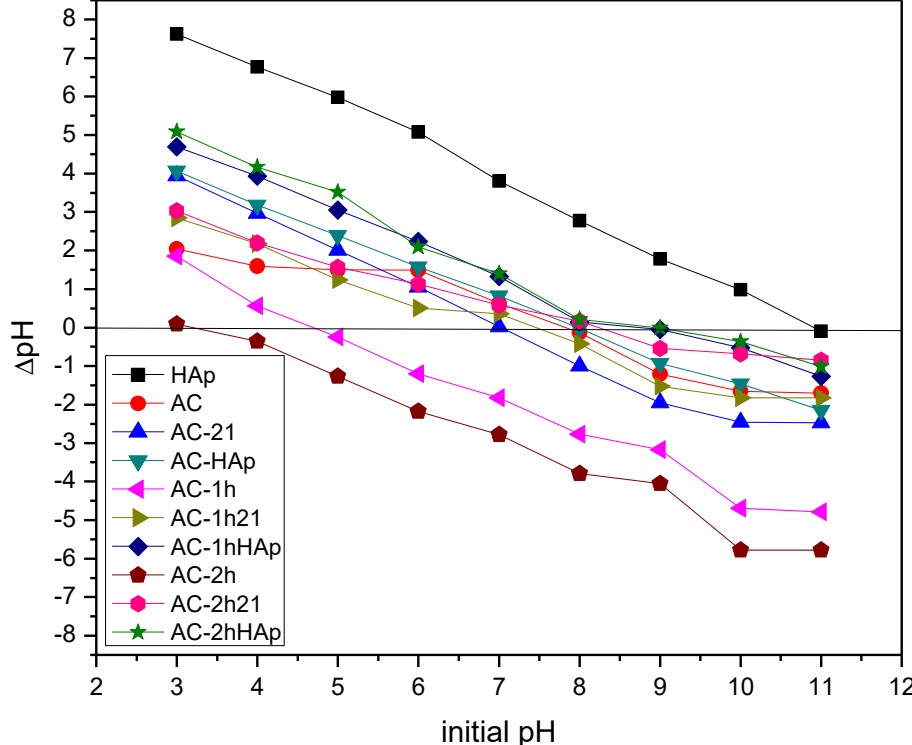

**Figure 8.** Charge distribution of the different materials.

For the commercial material (AC), a PZC of 7.82 with a relatively basic distribution was observed, and the value was affected by its contact with the $HNO_3$, which tended to acidify the surface based on the time the material spent with the oxidizing agent due to the oxygenated functional groups formed [15,34]. The AC-1h and AC-2h presented a PZC of 4.69 and 3.20, respectively [35].

The materials immersed in the SBF showed an increase in the PZC, with a tendency to be more basic, according to the charge distribution, due to the presence of ions from the SBF solution and hydroxyapatite deposits. The materials with only 21 days of incubation presented a basic behavior compared to the precursors (AC-1h and AC-2h), with a PZC of 7.48 and 8.28 for the AC-1h21 and AC-2h21, respectively. However, for the AC and AC-21 with PZCs of 7.89 and 7.01, a change in the PZC, determined by the insufficiency of the carboxylic sites that interact with the SBF ions, increased this value.

Finally, the materials with the addition of $CaSiO_3$ during incubation showed the highest PZC values of 7.98, 8.78, and 8.98 for the AC-HAp, AC-1hHAp, and AC-2hHAp, respectively, and correlated to the HAp deposits on the surface, which presented a basic behavior in the aqueous solution [36].

Batch adsorption experiments were carried out, and the corresponding isotherms were performed. Figure 9 presents the adsorption isotherms of the non-oxidized materials (AC, AC-21, and AC-HAp in Figure 9a), the oxidized materials for 1h (AC-1h, AC-1h21, and AC-1hHAp in Figure 9b), and the materials oxidized for 2h (AC-2h, AC-2h21, and AC-2hHAp in Figure 9c) in varying initial concentrations.

The materials presented very similar adsorption capacities, with an average removal percentage of 100% for the first two initial concentrations (0.5 and 1 mg·L$^{-1}$) and removal percentages of 96.78, 86.70, 73.24, 33.10, 16.21, and 8.22% for the concentrations of 2, 5, 10, 25, 50, and 100 mg·L$^{-1}$, respectively. The relationship between the concentration of the methylene blue in the solution with respect to the amount removed in the equilibrium made an L-I type isotherm for the adsorption isotherms [37]: it started with a linear behavior at the beginning and gradually reduced the adsorption capacity until the surface was saturated with the adsorbate, and the equilibrium was reached.

The parameters of the Langmuir and Freundlich models (Table 2) were calculated to estimate the adsorption capacity. An error (%), expressing the uncertainty of the measurements, is the ratio of the standard deviation divided by the average value of the measured constants. The average error value for the Langmuir model was 1.45%, and for the Freundlich model, it was 2.01%.

**Table 2.** Parameters of the adsorption models.

| Adsorbent | Langmuir Model | | | | Freundlich Model | | | |
|---|---|---|---|---|---|---|---|---|
| | $q_{max}$ (mg·g$^{-1}$) | $k_L$ (L·mg$^{-1}$) | $R^2$ | Error (%) | $n$ (mg·g$^{-1}$) | $k_F$ (g·mg$^{-1}$·min$^{-1}$) | $R^2$ | Error (%) |
| AC | 7.336 | 4.618 | 0.910 | 1.325 | 0.075 | 5.577 | 0.867 | 1.942 |
| AC-21 | 8.399 | 1.394 | 0.971 | 0.753 | 0.150 | 4.674 | 0.782 | 2.326 |
| AC-HAp | 6.446 | 2.106 | 0.891 | 2.024 | 0.107 | 4.315 | 0.875 | 1.622 |
| AC-1h | 7.752 | 5.018 | 0.960 | 0.993 | 0.117 | 5.061 | 0.743 | 2.427 |
| AC-1h21 | 8.846 | 1.598 | 0.920 | 1.521 | 0.096 | 5.710 | 0.658 | 2.095 |
| AC-1hHAp | 7.202 | 1.732 | 0.915 | 1.224 | 0.120 | 4.548 | 0.831 | 1.931 |
| AC-2h | 9.144 | 1.436 | 0.932 | 0.937 | 0.111 | 5.846 | 0.735 | 2.120 |
| AC-2h21 | 9.818 | 2.123 | 0.730 | 2.486 | 0.061 | 7.664 | 0.883 | 1.321 |
| AC-2hHAp | 7.959 | 1.148 | 0.985 | 0.846 | 0.160 | 4.223 | 0.793 | 2.285 |

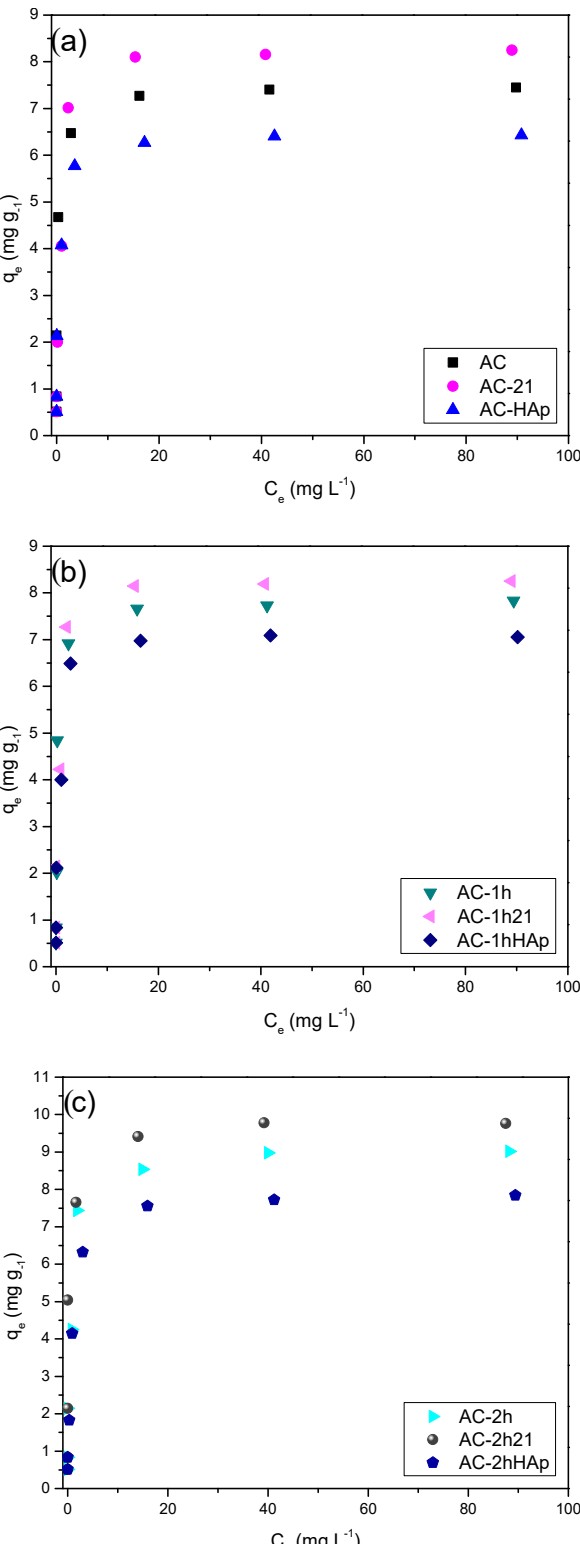

**Figure 9.** Adsorption isotherms of methylene blue on various carbon-based materials: (**a**) AC, AC-21, and AC-HAp; (**b**) AC-1h, AC-1h21, and AC-1hHAp; (**c**) AC-2h, AC-2h21, and AC-2hHAp, at 36 °C, pH 7, and initial dose of 0.1 mg of each adsorbent.

For the MB adsorption, most of the adsorbents fitted the Langmuir model, where a formation of a monolayer of the adsorbed molecules existed in this adsorption process [38]. All the superficial sites had the same possibility of being occupied but only by one molecule

of the adsorbate at the same time, and, in addition, an interaction between the adsorbed molecules was not observed [39]. The AC-2h21 was adjusted for the Freundlich model, but its $R^2$ value in the Langmuir was very close to this one; for this reason, the Langmuir model was used to describe the MB adsorption phenomena for all the adsorbent materials. The Langmuir estimated the maximum adsorption capacity ($q_{max}$) of the adsorbents, and an average value, as compared with other similar adsorbents, was observed [40–44], which increased with the $HNO_3$ treatment and the immersion of the SBF without $CaSiO_3$. Due to oxidation, the AC-1h and AC-2h showed a higher pore diameter, so a better interaction of the adsorbate molecule with the surface of the material was observed, with an increase from 7.33 $mg \cdot g^{-1}$ to 7.75 $mg \cdot g^{-1}$ and 9.14 $mg\ g^{-1}$. For the 21-day-incubated samples, the increase in the adsorption capacity occurred because of the PZC, AC-21 with a $q_{max}$ of 8.40 $mg \cdot g^{-1}$, AC-1h21 with a $q_{max}$ of 8.45 $mg \cdot g^{-1}$, and AC-2h21 with a $q_{max}$ value of 9.81 $mg \cdot g^{-1}$. For the adsorbents with HAp deposits in the pores, a decrease in the maximum adsorption capacity for the MB was observed (6.44, 7.20, and 7.95 $mg \cdot g^{-1}$, respectively, for the AC-HAp, AC-1hHAp, and AC-2hHAp).

Many mechanisms have been attributed to adsorbent materials, such as electrostatic attraction, ion exchange, the formation of hydrogen bonds, pore-filling, and $\pi$–$\pi$ interactions [45–48]. Some of the mechanisms reported were excluded due to the monolayer formation of the adsorbate molecules on the surface of the adsorbent and the micropores in the AC [49]. Therefore, the mechanism by which MB was adsorbed onto the materials is due to the $\pi$–$\pi$ interactions between the electron-$\pi$ of the carbonaceous adsorbents and the electron-$\pi$ of the aromatic ring of the adsorbate, according to the report by Tran et al. in 2017 [50].

A decrease in the adsorption capacity of the materials with the HAp deposits was observed due to the deposits impeding the correct $\pi$–$\pi$ interaction between the adsorbate and adsorbent at higher concentrations. Furthermore, for the adsorbents incubated without $CaSiO_3$, a combination of two mechanisms can explain the increased adsorption capacity: $\pi$–$\pi$ interactions and a possible charge/ion exchange interaction.

### 3.3. MB Adsorption Kinetics

To understand the adsorbents kinetic behavior, Figure 10 shows the experimental data. For all the adsorbents studied, the equilibrium time was longer. Due to the AC precursor (the coconut shell) the pore distribution was characterized by a high presence of micropores and a high surface area [35], and therefore, the equilibrium time was longer.

The experimental kinetic data were modeled using the pseudo-first-order and pseudo-second-order kinetic models, and the kinetic parameters, together with the $R^2$, are shown in Table 3. According to the value, the model that presented the highest correlation coefficients was the pseudo-second-order model, and the $q_e$ values were similar to those of the experimental data. The pseudo-second-order indicated that the adsorption rate was controlled by chemisorption, where the oxygenated functional groups, formed during the oxidation process, interacted with the MB molecule by sharing or electron exchange by valence forces [51].

**Table 3.** Parameters of kinetic models.

| Adsorbent | Pseudo-First-Order | | | | Pseudo-Second-Order | | | |
|---|---|---|---|---|---|---|---|---|
| | $q_e$ ($mg \cdot g^{-1}$) | $k_1$ ($min^{-1}$) | $R^2$ | Error (%) | $q_e$ ($mg \cdot g^{-1}$) | $k_2$ ($g \cdot mg^{-1} \cdot min^{-1}$) | $R^2$ | Error (%) |
| AC | 0.952 | 0.011 | 0.995 | 0.968 | 1.263 | 0.007 | 0.998 | 0.863 |
| AC-21 | 0.783 | 0.029 | 0.934 | 1.436 | 0.916 | 0.039 | 0.961 | 1.222 |
| AC-HAp | 0.736 | 0.020 | 0.920 | 1.231 | 0.870 | 0.027 | 0.945 | 1.507 |
| AC-1h | 0.779 | 0.032 | 0.932 | 1.392 | 0.893 | 0.048 | 0.957 | 1.689 |
| AC-1h21 | 0.710 | 0.073 | 0.796 | 1.929 | 0.780 | 0.136 | 0.880 | 1.058 |

**Table 3.** *Cont.*

| Adsorbent | Pseudo-First-Order | | | | Pseudo-Second-Order | | | |
| --- | --- | --- | --- | --- | --- | --- | --- | --- |
| | $q_e$ (mg·g$^{-1}$) | $k_1$ (min$^{-1}$) | $R^2$ | Error (%) | $q_e$ (mg·g$^{-1}$) | $k_2$ (g·mg$^{-1}$·min$^{-1}$) | $R^2$ | Error (%) |
| AC-1hHAp | 0.739 | 0.022 | 0.887 | 1.568 | 0.856 | 0.033 | 0.914 | 1.231 |
| AC-2h | 0.867 | 0.017 | 0.989 | 0.995 | 1.125 | 0.014 | 0.994 | 0.995 |
| AC-2h21 | 0.715 | 0.194 | 0.832 | 2.032 | 0.763 | 0.371 | 0.915 | 1.439 |
| AC-2hHAp | 0.754 | 0.028 | 0.750 | 1.986 | 0.794 | 0.074 | 0.829 | 1.802 |

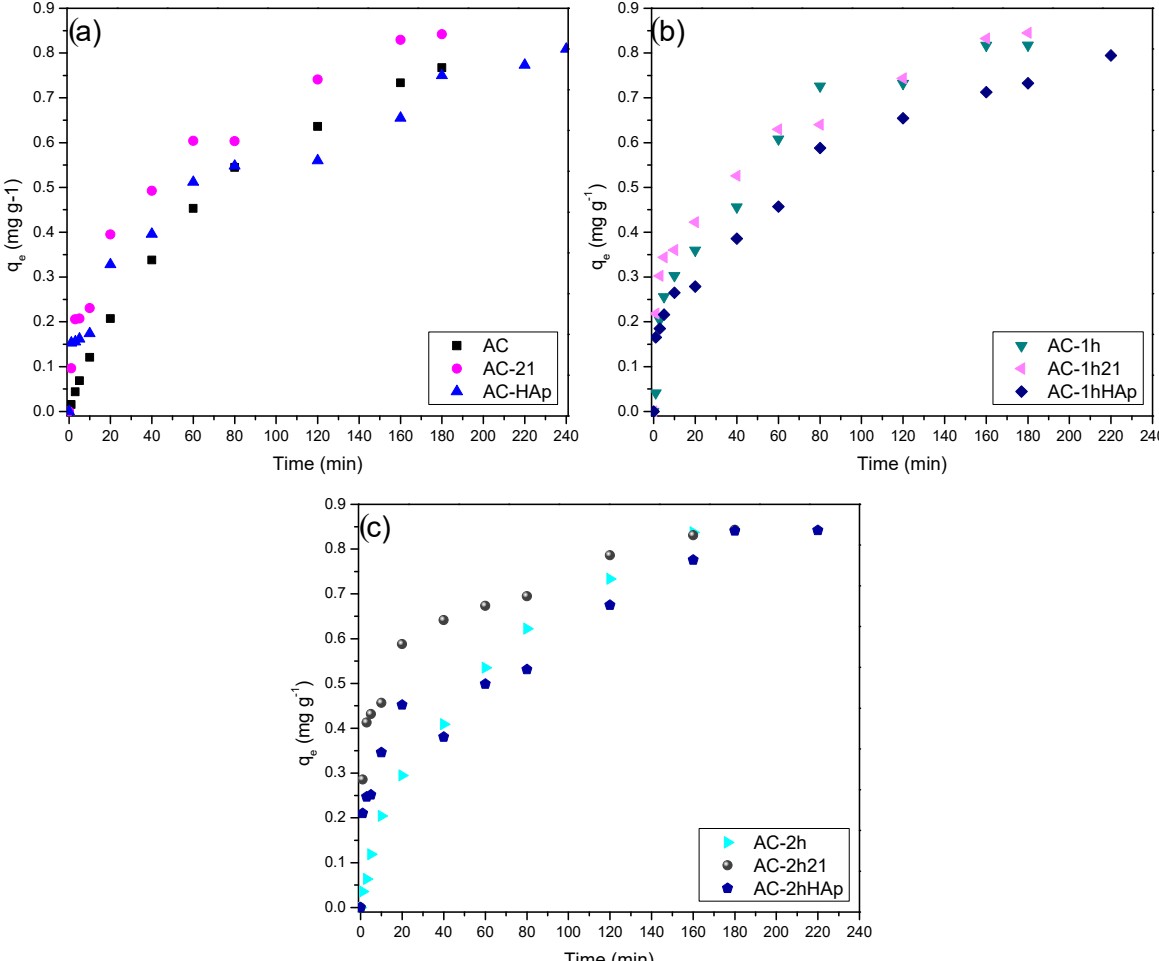

**Figure 10.** MB adsorption kinetics on different carbon-based materials (conditions: 1 ppm, pH 7, 60 mL volume, and 60 mg of the initial dose of each material). (**a**) Samples without oxidation, (**b**) samples with 1 h oxidation, (**c**) samples with 2 h oxidation.

The pseudo-second-order kinetics' parameters of the $k_2$ were directly related to the adsorption speed [52]. The $k_2$ increased according to the modifications in the AC, with the highest value of the $k_2$ at 0.37119 g·mg$^{-1}$·min$^{-1}$ for the AC-2h21, which also showed MB's highest adsorption capacity. In comparison with this material, the others showed a lower value of the $k_2$, and a longer time was necessary to reach the equilibrium. Based on the parameters $q_e$ and $k_2$ obtained from the kinetic model, the adsorption rate was calculated with the following equation (Equation (7)):

$$h = k_2 q_e^2 \tag{7}$$

Table 4 contains the values of $h$, and the highest value (0.2161 g·mg$^{-1}$·min$^{-1}$) was for the AC-2h21. The results depended on the initial concentration, providing a higher adsorption rate, and were confirmed with the values of the $k_2$ for how the global adsorption rate was.

**Table 4.** Parameters of kinetic models.

| Adsorbent | $h$ (mg·g$^{-1}$·min$^{-1}$) |
|-----------|------------------------------|
| AC | 0.012 |
| AC-21 | 0.033 |
| AC-HAp | 0.022 |
| AC-1h | 0.038 |
| AC-1h21 | 0.083 |
| AC-1hHAp | 0.024 |
| AC-2h | 0.018 |
| AC-2h21 | 0.216 |
| AC-2hHAp | 0.046 |

### 3.4. Effect of Contact Time on MB Adsorption

Figure 11 shows the effect of the contact time. The adsorption of the MB molecule was not as fast in general terms for all the adsorbents compared to previous works [41,53–55].

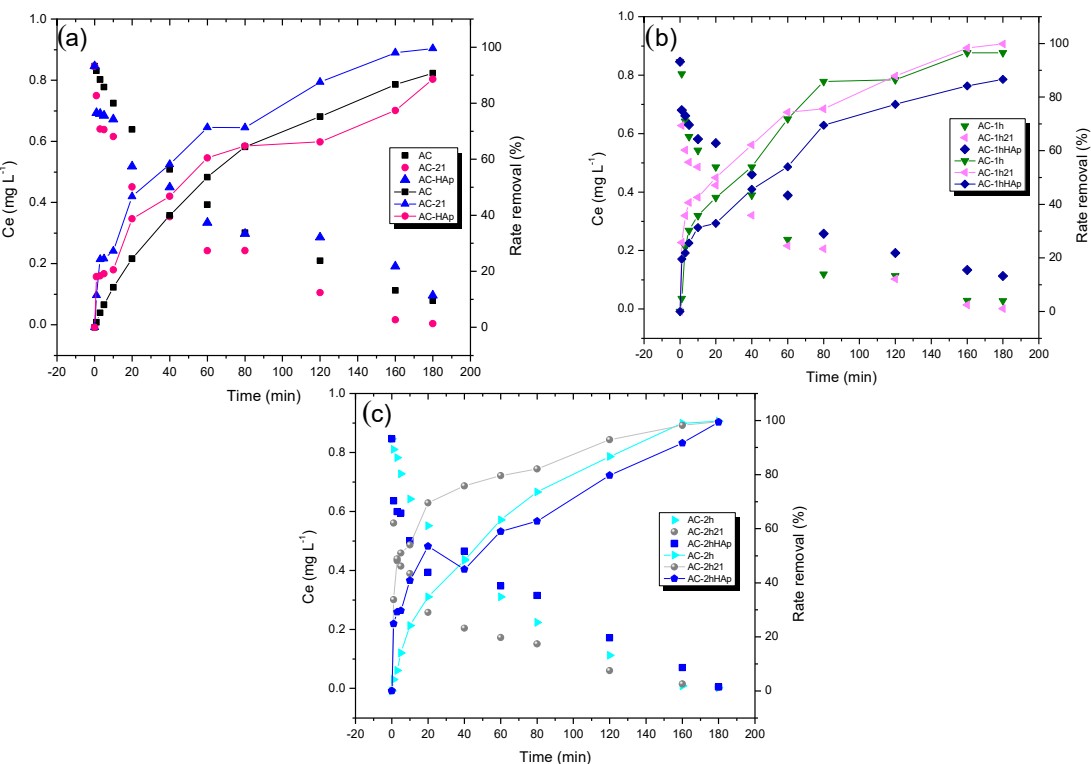

**Figure 11.** Effect of contact time as a function of the adsorbed concentration and removal rate of each material studied at experimental conditions of: 1 ppm, pH 7, and an initial dose of 60 mg. (**a**) Samples without oxidation, (**b**) samples with 1 h oxidation, (**c**) samples with 2 h oxidation.

The performance based on the contact time was described for all the adsorbents: at 1 min, the initial concentration of methylene blue decreased around 16.03%, the value continued to increase, and at 40 min, a remove percentage of 52.85% was observed; at 120

and 220 min, there was an average value for the removal of 82.21 and 98.31%. For this reason, it is possible to attribute that the minimum contact time for the adsorbents to remove 95.61% of the MB was 180 min. The removal rate was compared with the commercial adsorbent (AC), 90.7%, and modified HAp with the highest amount on the surface (AC-2hHAp), 99.4%. The results demonstrated that the modification with hydroxyapatite improves the removal rate at low concentrations of MB.

## 4. Conclusions

The oxidation time and the addition of $CaSiO_3$ were the key parameters for the growth of a greater amount of Hap, which was observed in the signals of the phosphate group in the FTIR analysis, in the activated carbon surface with agglomerated deposits of particles with a sphere-like morphology in the SEM microscopy, and in a higher amount of Ca and P in the EDS analysis. The ash content and XRF also confirmed that these parameters determined the HAp growth in the sample AC-2hHAp, with the highest oxidation time and the addition of $CaSiO_3$ during the incubation process. The charge distributions showed that the oxidized materials had a low PZC compared to that of the AC, and the incubated/modified with HAp showed a higher PZC due to the presence of hydroxyapatite with a basic behavior.

The MB adsorption capacity of the materials was better fitted to the Langmuir model, with a $q_{max}$ of 9.81 mg·g$^{-1}$ for the AC-2h21. In addition, the MB adsorption kinetics followed the pseudo-second-order reaction model, with an average removal of 95.61% of the MB at a contact time of 180 min in a concentration of 1 mg L$^{-1}$. The results demonstrated good adsorption behavior from the synthesized composites for organic molecules, such as dyes, even with the modification with hydroxyapatite with a removal rate of 99.4% in 180 min for the AC-2hHAp. The characteristics of the obtained materials with hydroxyapatite could be suitable for the adsorption process due to the ion exchange capacity of HAp and the surface of activated carbon.

**Author Contributions:** A.M.-S.: The student performed the experimental part and wrote the introduction and methods of the manuscript. J.C.R.-H.: designed the experimentation, followed up on the methodology, and wrote the results and discussion of the manuscript. S.E.F.-V.: supported in the experimentation and discussion of infrared spectroscopy characterizations. A.G.E.-G.: obtained the micrographs and discussed the results of the structural properties. J.Y.G.-C.: conducted the ash content experiments and supported the discussion of the results. F.P.L.-C.: supported the thermodynamic and kinetic experiments of methylene blue adsorption and analyzed the results together with the student. G.B.E.-I.: supported the experimental procedure for the obtention of the composites without a calcium silicate addition and the discussion of the results. All authors have read and agreed to the published version of the manuscript.

**Funding:** Anastasio Moreno-Santos received a grant from CONACyT. Grant number 1033316.

**Institutional Review Board Statement:** Not applicable.

**Informed Consent Statement:** Not applicable.

**Data Availability Statement:** Not applicable.

**Acknowledgments:** To Consejo Nacional de Ciencia y Tecnología (CONACyT) for the scholarship granted (grant number 1033316).

**Conflicts of Interest:** We know of no conflicts of interest associated with this publication, and there has been no significant financial support for this work that could have influenced its outcome.

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
