# Peer review of "Hydroxyapatite Growth on Activated Carbon Surface for Methylene Blue Adsorption: Effect of Oxidation Time and CaSiO3 Addition on Hydrothermal Incubation"

_applsci, doi:10.3390/app13010077_

Round 1

Reviewer 1 Report

I accept the present mansucript after major reviosion specifically in the charge distribution and PZC calcualtion portion. alos I would like to suggest to improve the following points.

1. there are some abbreavation used in the abstract which are difficult to understand for the reader, so please provid the full name.

2. I think the point of zero charge need to elaborate in more detaill. fig 8 y scale must be zeta potential so that can determine the pzc.

3. The author consider the reaction is the pseudo secodn order reaction need elaborat this.

4. conclusion part the first sentence is not clear. need to cleary state it.

Author Response

Authors are thankful for the comments that improves the manuscript in good ways, all the comments were responded as shown below:

Point 1: There are some abbreviations used in the abstract which are difficult to understand for the reader, so please provide the full name.

Response 1: The abbreviations for SBF, and characterization techniques were changed. 

Point 2. I think the point of zero charge need to elaborate in more detail. fig 8 y scale must be zeta potential so that can determine the pzc.

Response 2: We agreed with the reviewer, and we used the data to improve the PZC determination according to Mahmood et al in 2011 were the authors presented different methods for the obtention of PZC and no differences between the methods were observed, so we decided to rearrange the graph as the reviewer asked but we did not used zeta potential. Values obtained with this method presented a little variation of PZC, so changes in the values were made in the manuscript. However, the behavior of the obtained materials was like our results reported in the first submission.  The Figure 8 was changed according to the comment of the reviewer. We also added the reference of the method.

Point 3. The author consider the reaction is the pseudo second order reaction need elaborate this.

Response 3: We agreed with the reviewer, and we added the information of the mechanisms that a pseudo-second order model explains. We also added the reference that explains the mechanisms. Added to the manuscript: The pseudo-second order indicates that adsorption rate is controlled by a chemisorption, where the oxygenated functional groups formed during the oxidation process, interact with the MB molecule by sharing or electron exchange by valence forces. [51]

Point 4. Conclusion part the first sentence is not clear. need to cleary state it.

Response 4. We agreed with the reviewer, and we modified the sentence of the conclusion to better understand it. Added to the manuscript: Oxidation time and the addition of CaSiO3 were the key parameters for the growth of a greater amount of HAp, observed with signals of phosphate group in FTIR analysis, and in the activated carbon surface with agglomerated deposits of particles with a sphere-like morphology in SEM microscopy, with a higher amount of Ca and P in EDS analysis. Ash content and XRF also confirmed that these parameters determined the HAp growth in the sample AC-2hHAp, with the highest oxidation time and the addition of CaSiO3 during the incubation process.

Reviewer 2 Report

In the study, the authors investigated the formation process and mechanism of AC/HAp composites by AC surface oxidation and the addition of CaSiO3, and various characterizations. The adsorption performances of prepared composites for MB were also validated. Before accepted for publication, some issues should to be clarified.

1. The novelty need to be summarized both in the Introduction and Conclusion sections, e.g. since the adsorption performance of AC/HAp was inferior to that of AC, why the composites needed to be investigated?

2. The specific surface area of the prepared materials should be clarified to better demonstrate the adsorption performances and mechanism.

3. The removal rate (percentage) should be added into the Figure 11.

4. More literatures should be discussed about the adsorption of MB, such as Journal of Molecular Liquids, 263 (2018) 53-63; Molecules, 26 (2021) 6127; Water, 13 (2021) 2554.

Author Response

Authors are thankful for the comments that improves the manuscript in good ways, all the comments were responded as shown below:

Point 1. The novelty need to be summarized both in the Introduction and Conclusion sections, e.g. since the adsorption performance of AC/HAp was inferior to that of AC, why the composites needed to be investigated?

Response 1. We agreed with the reviewer, and we added the information required in the conclusions section, observed with the recommendation made in point 3. Is important to note that the material can be suitable for fluorine adsorption, however we are working in other manuscript about these experiments.

Point 2. The specific surface area of the prepared materials should be clarified to better demonstrate the adsorption performances and mechanism.

Response 2. Even though we agreed with the reviewer about the surface area and porosity, the measurements were not performed for this manuscript, since the micrographs demonstrated a good pore distribution, AC is obtained from coconut shell that presents a higher amount of micropores and the surface area is reported before. For the new materials the surface area will be measured later.  

Point 3. The removal rate (percentage) should be added into the Figure 11.

Response 3. We agreed with the reviewer and the removal rate was added to the graph in Figure 11.

Point 4. More literatures should be discussed about the adsorption of MB, such as Journal of Molecular Liquids, 263 (2018) 53-63; Molecules, 26 (2021) 6127; Water, 13 (2021) 2554.

Response 4. We agreed with the reviewer, and we discussed the mechanisms with the references recommended and we added these to the manuscript.

Reviewer 3 Report

The manuscript entitled "Hydroxyapatite growth on activated carbon surface..." written by A. Moreno-Santos et al., contains interesting and valuable results. I suggest its publishing after minor revision. The problems that should be solved in the manuscript are pointed out below:

Edition

1. The manuscript contains a small number of untypical expressions like:
(line 74) "HNO3 8 M" instead of "8 M HNO3"';
(e.g. 115, 116) "velocity" instead of "rater";
(e.g. 115, 116) "K" instead of "k" (rate constant is usually represented by a lowercase "k" letter);
(e.g. 121) "in triplicate" instead of "three times" (it just looks better, in my opinion...);
(139) "KW" instead of "kW";
(e.g. 207) "SEM microscopies" instead of "SEM images".

2. (162-164) The sentence is unclear to me. I suggest its rewriting.

3. (248) 779.00 (micro)m - Is it really so accurate? An "average" from the SEM image seems calculated from very different "spheres". Moreover, is it ~780 (micro)m? I see much smaller spheres in Fig. 6...

4. (248) "The addition of CaSiO3 in the incubation" - maybe "during incubation" will be more accurate?

5. (291) Why is there such a difference in the number of significant figures in pH values (7.82 and just 7)? It looks strange here.

6. Table II (not only this table) Why do some parameters in Table II have more and others have less significant figures, i.e. KF: 5.71029 but R2 = 0.638? A mean error of the average value is usually around 2-3%. I suppose it was similar here, so the significant figures should be reduced. Additionally, I strongly recommend the estimation of errors. A presentation of the calculated values without their errors is not professional (see also other Tables)...

7. (379) It would be much better to add units to the presented values during a discussion: "0.2161 g/(mg min)" instead of a lone "0.2161".

Other

8. (368-370) The second-order model usually gives better fitting parameters than the first-order one. The question is: has it got a physical sense? - It is just my comment for consideration.

9. (88) Is there any trick? It is extremely hard to adjust pH exactly to pH 7 using so concentrated NaOH and/or HNO3 (How to reach [H3O+] = 10-7 M using one million times more concentrated acid (0.1 M)?). Did you use any buffer, or was there another ingredient in the mixture that helped to do that?

10. (IR spectra in Fig. 2) What can be the reason for a little different shape of the IR spectra for AC-1h21 (a broad lower intensity ca. 2000 cm-1 and ca. 1100 cm-1)?

11. (267-268) Is the presence of calcium phosphate confirmed only by EDS or maybe also by IR or Raman spectroscopies, in that case?

Author Response

Authors are thankful for the comments that improves the manuscript in good ways, all the comments were responded as shown below:

Point 1. The manuscript contains a small number of untypical expressions like:
(line 74) "HNO3 8 M" instead of "8 M HNO3"';
(e.g. 115, 116) "velocity" instead of "rater";
(e.g. 115, 116) "K" instead of "k" (rate constant is usually represented by a lowercase "k" letter);
(e.g. 121) "in triplicate" instead of "three times" (it just looks better, in my opinion...);
(139) "KW" instead of "kW";
(e.g. 207) "SEM microscopies" instead of "SEM images".

Response 1. We agreed with the reviewer, and we changed the expressions in all the manuscript (see at the new manuscript attached).

Point 2. (162-164) The sentence is unclear to me. I suggest its rewriting.

Response 2: We agreed with the reviewer, and we changed the sentence in the manuscript: Inorganic materials remained in the crucible at the end of the analysis, since the boiling point is higher than the temperature used for ash determination.

Point 3. (248) 779.00 (micro)m – Is it really so accurate? An “average” from the SEM image seems calculated from very different “spheres”. Moreover, is it ~780 (micro)m? I see much smaller spheres in Fig. 6…

Response 3: We agreed and appreciate the comment of the reviewer, so we discussed more the SEM image (Figure 6c). We added to the manuscript: The particle size varies due to agglomeration, as particles of smaller than average size were visible (Figure 6c).

Point 4. (248) “The addition of CaSiO3 in the incubation” – maybe “during incubation” will be more accurate?

Response 4. We agreed with the reviewer, and we changed the word.

Point 5. (291) Why is there such a difference in the number of significant figures in pH values (7.82 and just 7)? It looks strange here.

Response 5. There was a change in the graph since a new method was recommend by another reviewer, so the values changed. We understand the comment and we changed the significant figures.

Point 6. Table II (not only this table) Why do some parameters in Table II have more and others have less significant figures, i.e. KF: 5.71029 but R2 = 0.638? A mean error of the average value is usually around 2-3%. I suppose it was similar here, so the significant figures should be reduced. Additionally, I strongly recommend the estimation of errors. A presentation of the calculated values without their errors is not professional (see also other Tables)...

Response 6: We agreed with the reviewer. We added the error (%) measured with the standard deviation and the average value for each model. Also we added the discussion how we obtained the error in the manuscript. We modified the significant figures to 3 in all the tables to adjust all the values in the manuscript.

Point 7. (379) It would be much better to add units to the presented values during a discussion: "0.2161 g/(mg min)" instead of a lone "0.2161".

Response 7. We agreed with the reviewer, and we added the units to improve the manuscript discussion.

Point 8. (368-370) The second-order model usually gives better fitting parameters than the first-order one. The question is: has it got a physical sense? - It is just my comment for consideration.

Response 8. We agreed with the reviewer, and we added a discussion about the pseudo-second order, in adsorbent materials commonly fits to this model due to the oxygenated functional groups and organic molecules, so if them are increased through oxidation process, the material will fit better to this model. 

Point 9. (88) Is there any trick? It is extremely hard to adjust pH exactly to pH 7 using so concentrated NaOH and/or HNO3 (How to reach [H3O+] = 10-7 M using one million times more concentrated acid (0.1 M)?). Did you use any buffer, or was there another ingredient in the mixture that helped to do that?

Response 9. We agreed with the reviewer about the difficulty to adjust the pH at 7, even all the samples in the laboratory were adjusted at the pH value mentioned with only a few drops of the HNO3 or NaOH, we added the accuracy that the potentiometer presents, and we modified the information in all the manuscript about pH adjustment. There is no buffer added, only the MB solution in water.

Point 10. (IR spectra in Fig. 2) What can be the reason for a little different shape of the IR spectra for AC-1h21 (a broad lower intensity ca. 2000 cm-1 and ca. 1100 cm-1)?

Response 10. We agreed with the reviewer, and we added a discussion about the intensity of the signals: An increase in the intensity on AC-1h21 for the phosphate group signal, can be attributed to the initial formation of HAp in the surface.

Point 11. (267-268) Is the presence of calcium phosphate confirmed only by EDS or maybe also by IR or Raman spectroscopies, in that case?

Response 11. We agreed with the reviewer, and we added the techniques that confirmed the formation of HAp on AC surface.

Reviewer 4 Report

Please label the IR spectra in Figure 2 with the associated functional groups.

Author Response

Authors are thankful for the comments that improves the manuscript in good ways, all the comments were responded as shown below:

Point 1. Please label the IR spectra in Figure 2 with the associated functional groups.

Response 1. We agreed with the reviewer and the functional groups were labelled in the Figure 2 and added to the manuscript.

Round 2

Reviewer 2 Report

In the revised manuscript, the authors have already made the corresponding changes for responding to my comments as frequently as possible. I have no additional comment. Therefore, I recommend this revised manuscript for publication in its current forms.